

# Interactive virtual fieldtrip as a tool for remote education.

Niki Evelpidou[1], Anna Karkani[1], Apostolia Komi[1], Aikaterini Giannikopoulou[1], Maria Tzouxanioti[1], Giannis Saitis[1], Evangelos Spyrou[1], Maria-Anna Gatou[1]

[1]Faculty of Geology and Geo-environment, National and Kapodistrian University of Athens, Panepistimioupoli, 15784, Greece

*Correspondence to*: Niki Evelpidou (evelpidou@geol.uoa.gr)

**Abstract.** Geoscience courses, such as geology and geomorphology, require not only classroom lessons and laboratory exercises, but field trips as well. However, the COVID-19 restrictions did not allow the prosecution of most planned field trips, and an alternative needed to be developed. The use of virtual field trips is one such alternative. Through them, one can not only visit any area of interest, but prepare themselves for any actual educational or exploratory field trip as well. Even though

they do not, and should not, substitute any physical visit of a site of interest, they have many advantages when combined with a "live" field work, in comparison to a field trip for which no preparation has been made.

Through this research, we compare the advantages and disadvantages of both virtual and real educational field trips based on the opinions of our students. We thus performed a virtual navigation on the island of Naxos, Cyclades (Aegean Sea, Greece) for a series of virtual field trips, which took place during webinars in the framework of Erasmus+ CIVIS. The virtual fieldtrip

was also presented to the third-year students of the Faculty of Geology & Geo-environment, National and Kapodistrian University of Athens, in the framework of the obligatory course of Geomorphology. Upon completion, all participating students were asked to fill in a questionnaire in order to evaluate the contribution of virtual field trips to their education regarding geomorphology and state their opinion as to whether they can supplement and/or substitute actual field trips. Most of them stated that virtual field trips can aid, but not substitute the actual field work. Most students mentioned that they would

attend another virtual field trip in the future, both as an alternative to classroom lessons and as a means of preparation for an actual field trip, but not in order not to attend the actual one.

## 1 Introduction

Geoscience courses, such as geology and geomorphology, require not only classroom lessons and laboratory exercises, but field trips as well. Through the latter, students can understand most of the principles taught in the classroom. Initially, students

can observe the Earth's relief, the landforms and other superficial features and they can gain a better understanding of the geological processes responsible for the shaping of the Earth's relief (Hurst, 1997). Moreover, students can co-operate with each other, as well as with their educators, and interact with nature and the environment, thus build a team spirit (Clark, 1996). In the field, one can recognize several geological structures and landforms and comprehend the processes that have led to their formation (e.g. Hurst, 1997).



However, the COVID-19 restrictions did not allow the prosecution of most planned field trips. Therefore, an alternative needed to be developed. The use of virtual field trips  one such alternative. Through them, attendants are able to observe a site in both two and three dimensions and from several viewpoints. One can visit any area of interest, regardless of its location, whereas no accessibility issues arise (Stainfield *et al.,* 2000; Carmichael & Tscholl, 2011). They can be attended by anyone, i.e. by both students, as well as educators, from all over the world (Stainfield *et al.,* 2000). Furthermore, photos, videos and maps can be

utilized, thus aiding the comprehension of the geological and geomorphological processes (e.g. Stainfield *et al.,* 2000). Among their advantages over actual field trips is the ease of their usage, as they can be attended by anyone who has a corresponding device, such as a computer, cellular phone or tablet, as long as it is connected to the internet (Çalışkan, 2011).

It is also worth mentioning that virtual field trips are exempt from several drawbacks of live field work. It is perceivable that, in order to visit an area, whether for recreational or investigative/educational purposes, several issues arise, such as the

transportation cost and time, as well as accessibility issues. For instance, is it very difficult for Greek educators to plan a visit to a remote area such as Russia or Germany, let alone prosecute it, as the time and cost of both the transportation and the accommodation would be exorbitant. The situation would be even more difficult for areas in other continents such as Africa and America. On the contrary, through the use of virtual field trips, one can navigate over any place of Earth, even in remote and/or inaccessible sites (Hurst, 1997). Additionally, neither safety issues (such as mountain climbing) nor weather conditions

(such as storms, wind or snow) obstruct the smooth prosecution of a virtual field trip, which is not always the case with live field work (Çalışkan, 2011).

Furthermore, virtual field trips have much more to offer than a simple presentation, as they are more interactive and interesting for most students, given that they have the ability to visit interesting sites themselves and perform an unofficial research, investigation and/or presentation (Gratton, 1999; Stainfield *et al.,* 2000). Another advantage of paramount significance is the

ability to prepare for a live field work, both educational and investigative. One can study for instance any observable superficial feature, such as the drainage network, the landforms, the coastline and the relief itself and make any geological and geomorphological notes before visiting an area of interest, as well as choose several sites to be visited when the actual field trip takes place (Gilmour, 1997; Stainfield *et al.,* 2000; Cliffe, 2017). This means that, even though they cannot or should not substitute an actual field work, virtual field trips can be utilized as a means of preparation.

As it has already been mentioned, field trips are essential for geoscience students. In the framework of Erasmus+ CIVIS, a European Civic University formed by the alliance of nine leading research higher education institutions across Europe, a series of virtual field trips were organized, amongst which was a virtual field trip in Naxos Island, Cyclades, Greece. The same virtual fieldtrip was also provided for the third-year students attending the mandatory course of Geomorphology at the Faculty of Geology and Geoenvironment of the National and Kapodistrian University of Athens (NKUA), since the restrictions due to

the COVID-19 disease did not allow the realization of any of the initially planned field trips. We prepared a virtual navigation in Naxos, an island of the central Cyclades, an area that is usually not selected for field trips due to its remoteness and its inaccessibility by car, as well as the consequent cost. The main aspects addressed in this virtual field trip are coastal evolution, palaeogeographic reconstruction, sea-level changes and geoarchaeology. It was initially developed as an alternative of the field





trips that cannot be performed, but we also tried to evaluate virtual field trips in general, both as supplements and substitutes
of live field work. Therefore, after its completion, students were asked to evaluate the virtual field trip and their general experience. Our ulterior goal was to give prominence to the significance of virtual field trips as supplements of actual field work and their insufficiency as substitutes.

## 2 Materials and Methods

The virtual field trip was designed using the ArcGIS StoryMaps application. Story maps are web applications containing
interactive maps, which are usually accompanied with texts in the form of narrative, as well as images, videos and links. StoryMaps enable the presentation of an area or event, in an interactive way, providing both geographic and descriptive data. Data were initially developed and organized using ArcMap v. 2.6. The GIS data included geological and geomorphological information, as well as palaeogeographic maps, relative sea level changes and geoarchaeological studies. Afterwards, the main narrative for the virtual fieldtrip was designed. Using ArcGIS StoryMaps, different tabs/chapters were developed containing a
number of thematic maps, which are accompanied by texts, videos and drone photographs, as well as external links to documentaries on the features presented (Fig. 1). The developed StoryMap is available here.

In order to provide a more comprehensive overview of the study area, a field trip guide was also composed, where one can find not only general geographical, physiographical, climatic, geological and geomorphological information about the island, but also information about the individual landforms. This information regards their geographical distribution over the island
and their formation processes in connection with the geomorphology and geology of the contextual study areas. Consequently, the virtual field trip was presented at 84 students for the CIVIS presentation and 134 third-year students attending Geomorphology. 94 students completed an anonymous questionnaire created for this purpose.



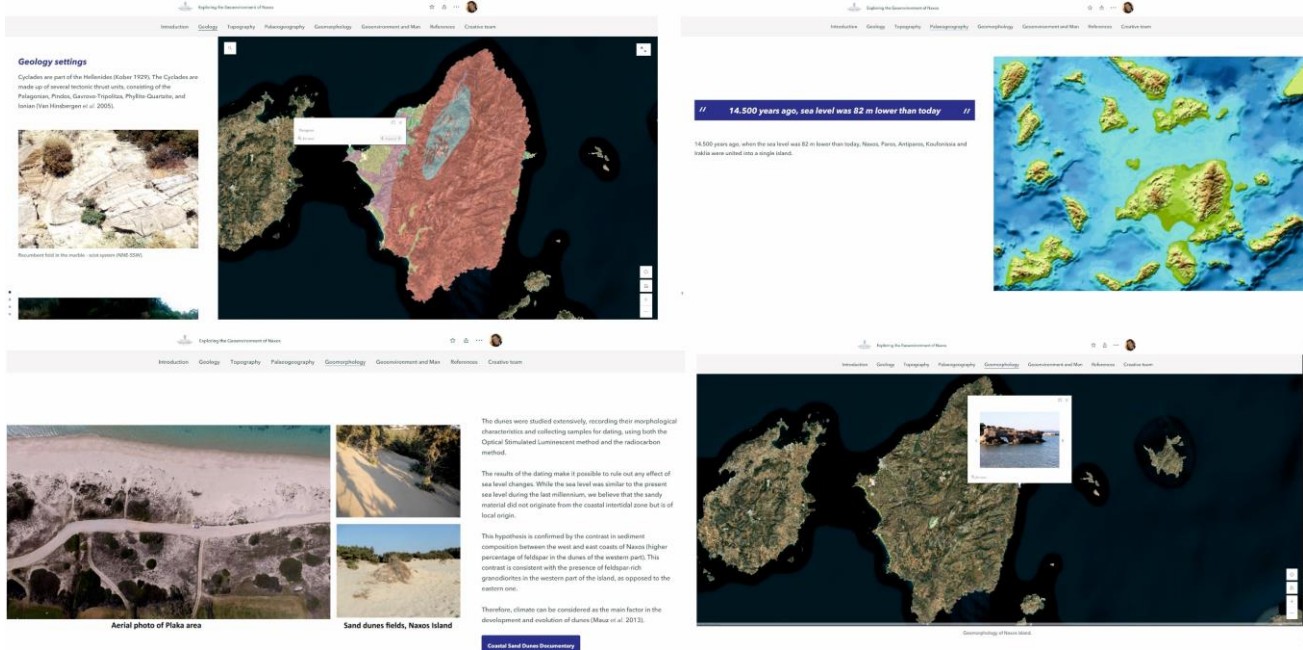

**Figure 1: Examples of the virtual fieldtrip developed using ArcGIS StoryMaps.**

## 3 Study area

Naxos is located in the central Aegean Sea and belongs to the south Cyclades complex (Fig. 2). It is the largest island of the Cyclades, covering an area of approximately 428 $Km^2$. The climate type of the area is the Mediterranean, i.e. dry and long summers and mild winters. Storms are common during winter, whereas snow is very rare.

### 3.1. Geodynamic setting

The Aegean is a unique environment, regarding the active geodynamics, ongoing geological processes and dynamically 90 changing landscapes due to the convergence between the Eurasian and African continental plates. The broader area of the Cyclades is a geotectonically active region, due to this convergence (e.g. Sakellariou & Galanidou, 2016). The seismicity of the Central Aegean region is, however, not high, as large earthquakes are rare (e.g. Papazachos, 1990; Leonidopoulou, 2008). The island of Naxos belongs to the Cycladic plateau (e.g. Gaki-Papanastassiou *et al.*, 2010), in the back-arc basin of the Hellenic arc. This has resulted in the complicated tectonic and geodynamic structure of the island of Naxos (Le Pichon & 95 Angelier, 1979; Angelier *et al.*, 1982; Le Pichon, 1982; Le Pichon *et al.*, 1982, 1995; Meulenkamp *et al.*, 1988; Taymaz *et al.*, 1991; Jolivet, 2001)..



The coasts of Naxos island, especially since Late Pleistocene, are determined by eustatic sea-level changes, whereas vertical tectonic movements bear no significant impact (Sakellariou & Galanidou, 2016); however, there are indications of tectonic control for the late Holocene in the broader area (Desruelles *et al.*, 2009; Lykousis, 2009; Evelpidou *et al.*, 2014). Evelpidou

*et al.* (2014), for example, have found several submerged tidal notches in the southeastern Cyclades, which were attributed to former seismic events, which have taken place since 3300 BP (Evelpidou et al. 2014, 2018). It is worth mentioning that no evidence of tectonic uplift (such as uplifted terraces, notches or beachrocks) has been found. There are many submerged landforms and other indicators at Naxos, such as beachrocks found down to a depth of 6 m (Evelpidou *et al.*, 2012; Karkani *et al.*, 2017).

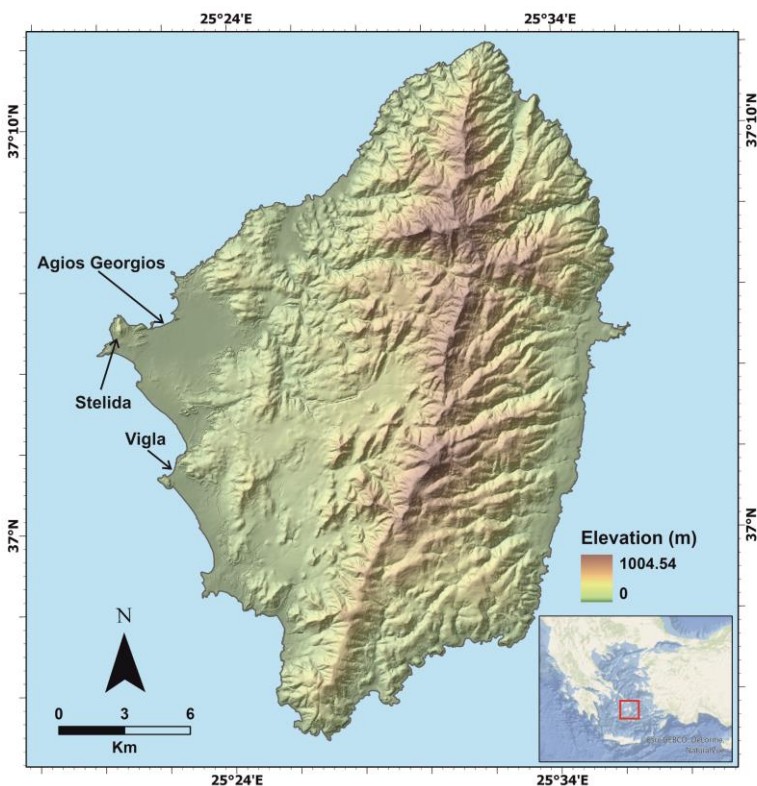

**Figure 2: Location of the study area and the sites discussed in the text (Created using ArcGIS Pro).**


### 3.2. Geological setting

Several authors (e.g. Trikkalinos, 1947; Dürr *et al.*, 1978; Robertson & Dixon, 1984; Papanikolaou, 2015) have subsumed the island of Naxos in the Attica-Cyclades massif, consisting of the non-metamorphic Cycladic Unit and the metamorphic Southern Cyclades Unit (e.g. Papanikolaou, 2015). The former mainly covers the northwestern part of the island. The latter

consists of a pre-Alpine basement including gneisses, amphibolites and marbles of Permian-Triassic age, underlying neritic



marbles and metabauxites, probably of Mesozoic age, their contact being a detachment (Jansen & Schuiling, 1976; Dürr *et al.*, 1978; Henjes-Kunst & Okrusch, 1978; Henjes-Kunst & Kreuzer, 1982; van der Maar and Jansen, 1983; Feenstra, 1985; Andriessen *et al.*, 1987). The metamorphic rocks (marbles, schists, metapelites etc.) surround a characteristic elliptical migmatite dome located in the center of the island. These units are disrupted by an Upper Miocene granodiorite, mainly

covering the western part of Naxos (Jansen and Schuiling, 1976). The western part of the island is also characterized by the presence of Neogene and Quaternary sediments (Evelpidou, 2001).

The lithology of Naxos consists of various lithological formations with different resistance in erosion. These differences, in combination with the action of the hydrographic network and tectonics, have determined the present morphology. Evelpidou (2001) has identified six lithological units, namely marbles-schists, the migmatite, the granodiorite, Neogene sediments,

Quaternary sediments and colluvial-alluvial sediments. Marbles and schists cover the largest part of the island, namely the central, northern, southern and eastern part, except for the migmatite dome in the center. Neogene, Quaternary and colluvial – alluvial sediments are almost absent in this part of the island and, alongside the granodiorite, they are abundant on the west coasts of Naxos (Evelpidou, 2001).

### 3.3. Geomorphology

The relief of Naxos is mainly mountainous, especially its eastern, northern and southern part. It is characterized by a mountain range with a direction N–S. Its highest summit, Zeus, is found in the center of the range and reaches 1,001 meters, which is also the highest point of the Cyclades. The western part of the island is, on the contrary, milder (Evelpidou, 2001). As already noted, the lithology of Naxos consists of various lithological formations with different resistance to erosion, which, in combination with the action of the hydrographic network and tectonics, has created the present morphology. Therefore, Naxos

has many landforms and geomorphological features that result from differential erosion (Evelpidou, 2001). This geological and tectonical setting has led to the formation of several landforms, associated with several environments (e.g. coastal, lagoonal, aeolian etc.) (Evelpidou, 2001). This is the main reason why this region was chosen as a study area and as a virtual field trip destination. It has the potential to aid students in recognizing various landforms, whereas it can offer them information about their formation processes. Additionally, several sites have an intriguing palaeogeographical evolution during the

Quaternary, which can further aid students in understanding the principles of geomorphology.

Due to the variation of lithologies, the coasts of Naxos are also diverse. The western part, as mentioned before, is mainly dominated by alluvial sediments and granodiorite, hence the low morphology and the abundance of beaches. On the contrary, the rest of the island's coasts include other lithologies (marbles and schists) and are relatively steep and with very few beaches (Evelpidou, 2001).





**3.4. Landforms**

**3.4.1. Tafoni**

Tafoni are arc-shaped weathering cavities mainly formed in crystalline rocks (such as granites) of medium to large granules, but they can also be found on other rocks, such as sandstones, limestones and shales. Their width and depth vary from a few centimeters to several meters (Turkington 2004; Migoń & Maia, 2020) while their shape tends to be ellipsoid or spheroid.

Their origin is related to the action of the wind and especially with the sea-water spray.

Although there are many tafoni sites over Naxos, the most characteristic tafoni and cavernous weathering formations appear in Stelida area, developed on granodiorite (Evelpidou *et al.*, 2021a). The cavernous weathering forms are found 25 m above mean sea level. Evelpidou *et al.* (2021a) have studied roughly 200 tafoni and honeycombs, regarding their geomorphological characteristics, mineralogy and geochemistry. Taking into account that the majority of the investigated honeycombs and tafoni

present rock fragments in the form of debris and flakes, it appears that their evolution is still active (Evelpidou *et al.*, 2021a). According to the analysis of their morphological characteristics, their evolutionary stage corresponds to the coalescence stage that is characterized by wall breakdowns. The increased presence of micas and gypsum along with the presence of halite at littered rock fragments suggests that salt weathering, due to sea moisture and seawater spray, has played an important role in the development of these cavernous weathering forms (Evelpidou *et al.*, 2021a). Furthermore, biotite alteration may be a key

factor not only for the tafoni formation, but for the yield of iron bearing bacteria in the cavities as well (Evelpidou *et al.*, 2021a). The growth of iron rich bacteria in between biotite flakes may contribute to mineral break down producing more cracks and cavities on the rock surface.

**3.4.2. Sand dunes**

Sand dunes are common landforms along the western beaches of Naxos (e.g. Evelpidou *et al.*, 2010, 2012; Cordier *et al.*,

2011). They are usually developed in front of coastal lagoons, part of which lie below sea level. The sand dunes of Naxos are very important, not only due to their function as habitats, but also because they play a protective role against coastal flooding due to sea-level rise. Amongst the most characteristic dune fields in Naxos are those with cedar species (Fig. 3), belonging to the Natura 2000 protected areas. Morphological investigations and OSL dating on the coastal dunes of Naxos have revealed that the dune formation occurred during the last millennium, while sands have a local origin, and should not be related to the

sea level change (Cordier *et al.*, 2011). In contrast, climate has largely influenced the evolution of dunes, both directly (role of wind) and indirectly (influence on vegetation and fluvial streams carrying sediments to the coast system) (Cordier *et al.*, 2011).





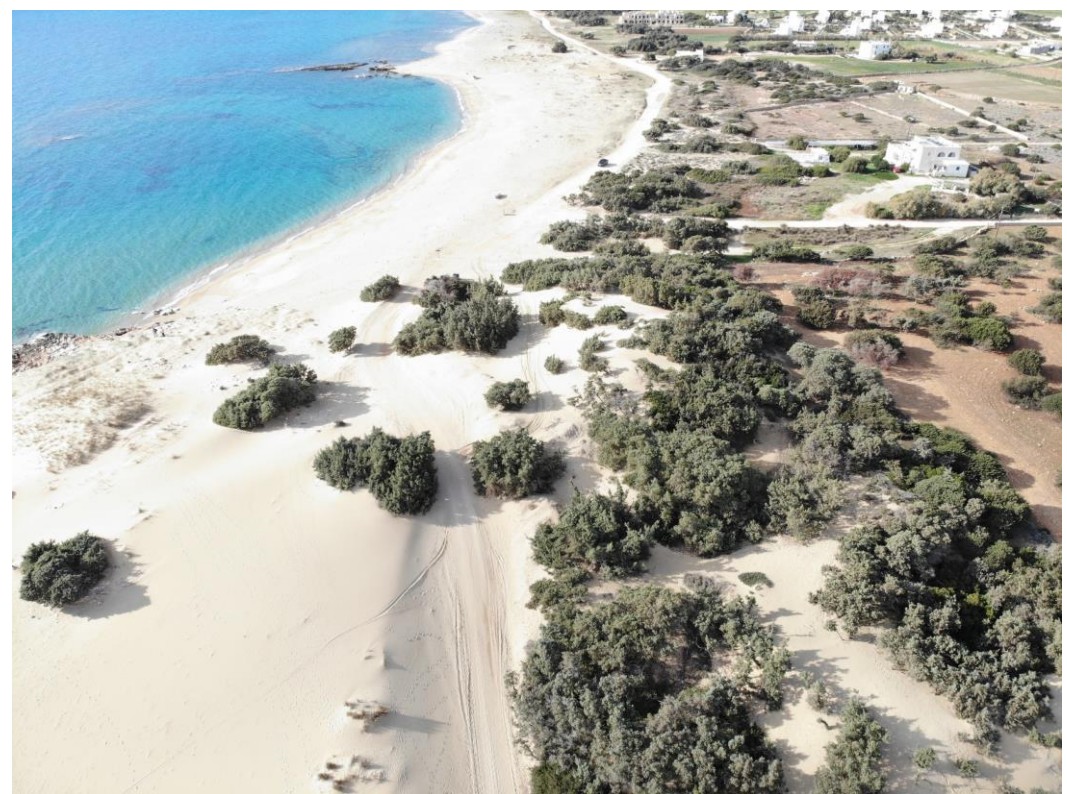

Figure 3: Characteristic dune fields in Naxos with cedar species.

### 3.4.3. Coastal wetlands and lagoons

Coastal wetlands and lagoons are particularly vulnerable to local palaeo-environmental changes and they constitute a powerful
tool for tracking changes of the last thousand years in the coastal zone. Sea-level changes, climatic changes and human
interventions are recorded in the sediments of coastal swamps and wetlands, making these environments ideal for
palaeogeographic reconstructions.

In Naxos island, many lagoons have been formed in the western coastal zone. Two characteristic examples highlight the
evolution of the western coasts during the late Holocene, Agios Georgios and Mikri Vigla (Evelpidou *et al.* 2010; 2012). At
Agios Georgios, a series of coring revealed that the area was a lagoon 6000 years ago while today it is a wetland. This area
used to be a harbor, as it was an active lagoon from at least 6144 BP until 232 BP, and probably served Yria at that time
(Evelpidou *et al.*, 2012). Nowadays, a tombolo formation is visible in the area. Its development is owed to coastal currents that
transported and deposited the sand between Manto island and the adjacent beach. The reef in front of Manto island consists of
beachrocks that lie slightly lower than sea level, down to a depth of 1.9 m. The underwater beachrocks represent fossilized
shorelines of 1500 years ago and 3400-4500 years ago (Karkani *et al.*, 2017). Today these beachrocks protect the coast from
erosion due to wave action, and sea level rise.




A few kilometers to the south, Vigla area is characterized by low land morphology crossed by torrents of temporary flow, coastal lagoons, dunes and sandy beaches (Fig. 4). Angular pieces of granodiorite and river-torrent depositions appear on the fringes of the alluvial plain. In the Vigla area, two beachrock benches, along with sand dunes represent the dominant coastal

landforms. The analysis of sediments and micro-faunal content from Vigla corings revealed that during 3800 BP to 1625 BP the area of Vigla was an active lagoon (Evelpidou *et al.*, 2012). At least during the Bronze Age (3300 BC-1200 BC), Vigla may have been a place where boats could anchor. The beachrocks acted as a "barrier" subsiding periodically, allowing the entrance of sea water into the gulf. Vigla area was protected, like St. Georgios Bay, by a beachrock bench. At the onshore section, the hydrographic network 3000 years ago probably discharged in the Vigla bay, while today part of this network has

migrated southeast (Evelpidou *et al.*, 2012).

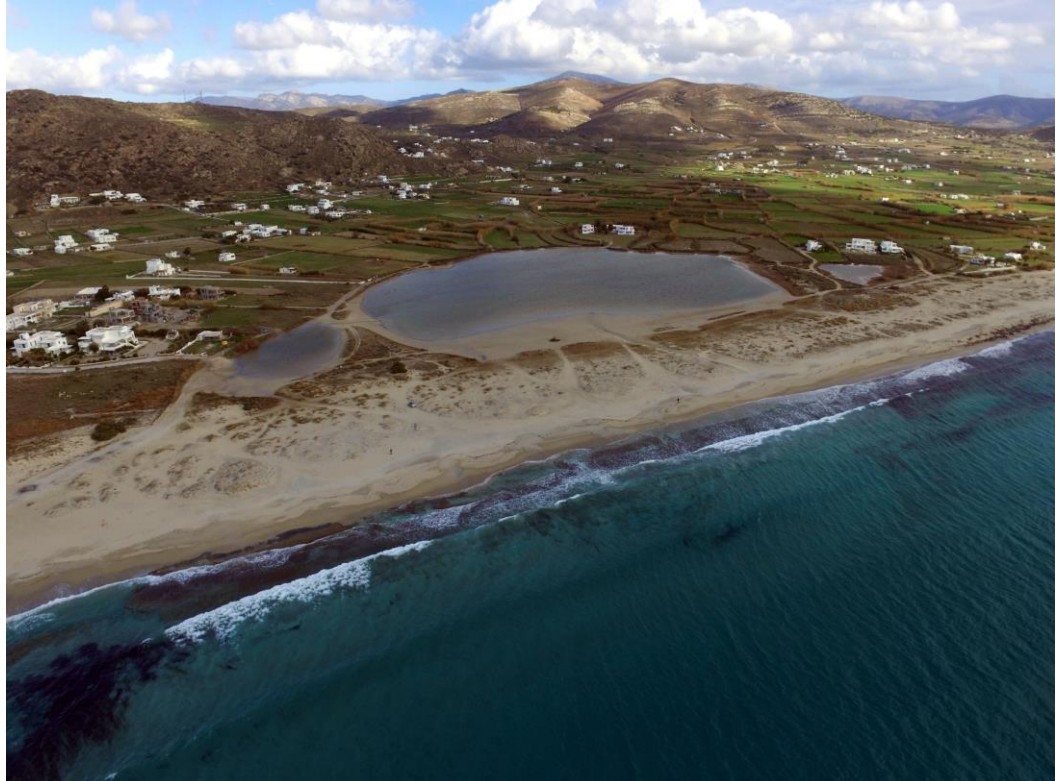

**Figure 4: The coastal zone of Vigla area.**

### 3.4.4. Tidal notches

Marine notches are horizontal U-shaped rock incisions along coastal cliffs. They are created through wave erosion, chemical

weathering and/or bioerosion. Tidal notches are a special type of marine notches, which form through bioerosion by several endolithic and epilithic organisms that reside near the mean sea-level and are mainly found on carbonate cliffs (e.g. Pirazzoli,





1986; Evelpidou & Pirazzoli, 2015). They form near the mean sea-level, which means that they can offer information about palaeo-sea-level changes, if found uplifted or submerged (e.g. Kershaw & Guo, 2001).

In Naxos, there are several sites where tidal notches are found, primarily in the NE and NW coasts, reaching depths of up to 2.8 m (Evelpidou *et al.*, 2013, 2014). Evelpidou *et al.* (2014) have identified seven submerged tidal notches in several islands of the central Cyclades, including Naxos, at depths of between $280 \pm 20$ and $30 \pm 5$ cm below the present sea-level. The vertical sequence of these submerged notches indicates rapid subsidence events, probably of seismic origin (Evelpidou *et al.*, 2014). The fossil shorelines were correlated with corings and archaeological information from the area in order to define their chronology. The subsidence of the uppermost notch at -30 ± 5 cm was attributed to the global sea-level rise during the last two

centuries and, at least partly, to the 1956 Amorgos earthquake (Evelpidou *et al.*, 2013). The deeper coastlines are probably owed to repeated subsidence events and not to gradual subsidence (Evelpidou *et al.*, 2014). These subsidence events have also affected the ancient coastal settlements at Naxos, who were probably forced to build their house floors higher and placed pebbles under their floors as a means of drainage, preventing the moisture to penetrate into the room (Evelpidou *et al.*, 2018).

### 3.4.5. Beachrocks

Beachrocks are common coastal landforms formed through the rapid cementation of beach material, the cement usually being aragonite and high-magnesian calcite (Bricker, 1971). The beach's material that is cemented can vary from fine to coarse coastal sediments, as well as artificial material (e.g. Bricker, 1971; Milliman, 1974; Vieira & Ros, 2007). Beachrock cements are typically calcitic or aragonitic, depending on the physicochemical factors of the diagenetic environment. The cement's mineralogy and morphology are indicators for the diagenesis environment (Mauz *et al.*, 2015).

In western Naxos, a series of extensive submerged beachrock slabs may be found, reaching depths of about 6 m. Detailed underwater surveys, analysis of aerial photogrammetry, microstratigraphic analysis and luminescence dating allowed the reconstruction of palaeogeographical changes, both spatially and chronologically and determine the relative sea-level changes during the late Holocene (Karkani *et al.*, 2017). The multiple analyses of beachrocks, sediment coring and archaeological data suggested that the relative sea-level rose by at least 3.8 m in the last 4.0 Ka and that relative sea-level variations in the last 2.0

Ka did not exceed 2 m with respect to the present mean sea-level (Karkani *et al.*, 2017).

### 3.5. Coastal erosion

Although there are many places around in the globe that are facing coastal erosion, in Naxos this phenomenon is rare, even though it has a very sensitive coastal zone, with several sites with unstabilised sand dunes and a lagoon behind. The reason is that beachrocks are developed parallel to the coastline, in most of the coasts, acting as a protective agent against wave action,

for the sensitive coastal zone. The site of Agios Georgios provides a clear view regarding the role of beachrocks as natural breakwaters, i.e. a natural protection for thousands of years, which inspired the experimental research for artificial beachrocks (Imran *et al.*, 2019; Saitis *et al.*, 2021). Sand, as well as bacteria, from the corresponding beach are used for the development of artificial beachrocks in the laboratory, using the microbially induced carbonate precipitation (MICP) technique. The





developed artificial beachrocks were afterwards studied, regarding their physical properties, estimating the Uniaxial

Compressive Strength (UCS), whereas a solidification test, a mineralogical study in a polarized microscope SEM-EDS and an X-ray tomography were conducted (Saitis *et al.*, 2021). Research so far has shown that the artificial beachrocks have similar properties to the natural ones, and that both natural and artificial ones, can ensure the protection of a coast from wave activity and the moderation of coastal erosion.

## 4. Results

Amongst the students who attended the CIVIS virtual field trip and the mandatory course of Geomorphology, 94 filled in the questionnaire (Fig. 5). The vast majority (90.4%) found the overall trip interesting and were aided in understanding the principles of Geomorphology. An 81.9% also found that there was adequate connection with other courses aside from Geomorphology. 89.4% stated that the means of this field trip was satisfying, given the COVID-19 restriction. A few of them (44.7%) had attended another virtual field trip before. It is worth mentioning that 35.1% of those who had attended another

virtual field trip before were the students of Geomorphology, given that another virtual field work had taken place shortly before this one (Evelpidou *et al.*, 2021b). Almost all of the students (93.6%) would attend another virtual field trip in the future, even if the restrictions are abolished, as trainees and in order to prepare themselves for an actual field trip. The main reasons for this are, according to them, the ability to view satellite images of previous time periods (62.8%), to panoramically view an area (56.4%), the easy usage/performance (47.9%), the ability to view areas in three dimensions (42.6%), the fact that virtual

field trips are time (40.4%) and money saving (38.3%) and other reasons. 91.5% of the students would use VFs in order to prepare themselves for actual field work, whereas 86.2% would use virtual field trips, as educators, instead of other common teaching methods such as power point presentations.

The students were asked to freely comment on the specific field trip, as well as virtual field trips in general, as opposed to actual field trips. 31 of them (33.0%) commented and mentioned several drawbacks of virtual field work. The most frequently

mentioned ones included the difficulty/inability to understand the principles taught through a virtual field trip as opposed to an actual one (19.4%), the limited interaction between students, educators and/or nature (16.1%), the inability to observe superficial features (e.g. landforms) in detail (16.1%), the inability to collect samples (e.g. of rocks) and/or perform *in situ* measurements (16.1%). 25.8% of them stated that virtual field work does not provide them with the knowledge, experience, skills, familiarization with the field etc. that they are going to need as future geologists. Finally, when asked if virtual field

trips can substitute actual ones, only 17 out of the 94 students (18.1%) answered positively.



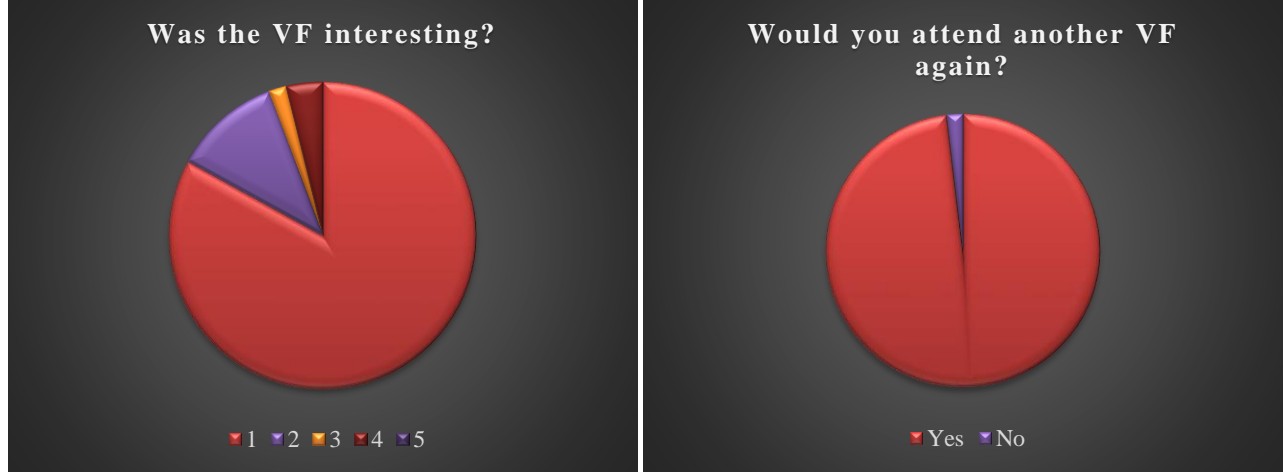

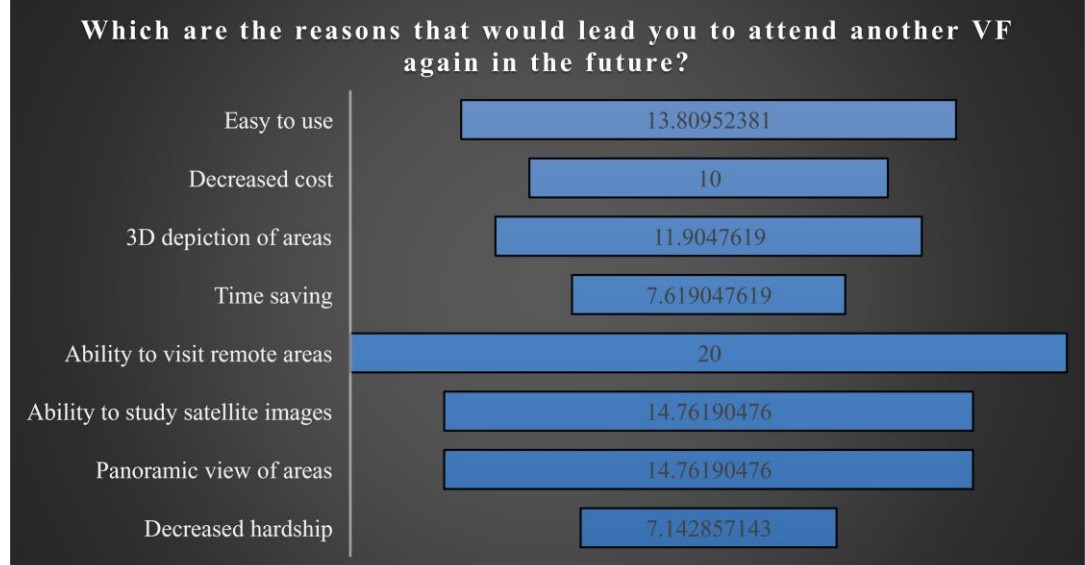

**Figure 5: Characteristic answers from the questionnaire by the students. The answers were selected from a list.**

## 5 Discussion

One of the main purposes of this virtual "excursion" was to train our students on coastal evolution, palaeogeographic reconstructions, sea-level changes and geoarchaeology. Therefore, through this virtual navigation, we intended to show the

students some of the main landforms of the island of Naxos and correlate them with the geological and geomorphological setting of the contextual area of interest. Through the ArcGIS StoryMaps application, they had the ability to observe geological, geomorphological and relief maps, photos, satellite images etc., thus combining the optical factor with their previous geological knowledge and the lectures during the virtual field trip. An important aspect of StoryMaps is the availability of interactive





maps. Students are able to freely interact with all the available maps within the developed virtual fieldtrip and obtain geographic and descriptive information for each site as well as multimedia material (i.e. photographs).

As mentioned above, most students were intrigued by this virtual field trip and were aided in understanding the principles of the course of Geomorphology. Additionally, they found it to be a useful way in preparing themselves for a live field work. Yet, as most of them mentioned, virtual field trips can only be used supplementarily and not as substitutes for physical field trips. After all, this is, or should be, how virtual field trips are used (e.g. Gilmour, 1997). They can, however, offer students a more interactive experience than other commonly used educational methods (such as power point presentations), that is they should be used in the classroom as an alternative teaching method (e.g. Ramasundaram *et al.*, 2005; Cliffe, 2017).

Virtual navigations are a very useful tool, offering many abilities when preparing a physical visit to an area. Initially, one can determine the sites of interest which are to be visited for sampling collections, filed measurements etc. Furthermore, one can use them to create maps, which can later be corrected, once the said sites have been visited. That can also happen vice versa, i.e. through a virtual field trip, existing data from the field can be reconsidered. What is more, in that way, remote and/or dangerous sites such as cliffs can be safely visited (Stainfield *et al.*, 2000) and most regions can be panoramically viewed, whereas persons with mobility limitations can also participate in virtual navigations (Hurst, 1997; Gilley *et al.*, 2015).

However, they cannot act as a substitute for physical field work (e.g. Gilmour, 1997), and they have several limitations for students (Spicer & Stratford, 2011; Mead *et al.*, 2019). They are deprived of the interaction with each other, with the educators and, mainly, with nature, whereas this would not be the case for live field trips (Clark, 1996; Gilmour, 1997). Even when the former are achieved, interaction with nature is restrained. Students cannot, for instance, use all their senses, namely vision, hearing, smell, touching, or even tasting, as the main purpose of a field trip is this interaction (e.g. Çalışkan, 2011; Han, 2020). Another aim of field trips is usually the understanding of the principles of geoscience courses taught in the classroom. This can be achieved to a greater extent in the field than in the classroom. Students can be greatly aided in understanding the geological, geomorphological and other natural processes when in the field (Hurst, 1997), while a field experience can be recalled and remembered by most students, as opposed to a classroom lesson. Furthermore, the students can neither collect rocks and other samples nor capture the landscape, the landforms and other features, when attending a virtual field trip. It is also worth mentioning that, even when the panoramic or three-dimensional view of an area is feasible, several features cannot be observed in detail, primarily small-scale ones, such as minerals, micro-fissures, bedding, karst cavities, granulometry etc.

**6 Conclusions**

Virtual field trips are a very significant tool for field work preparation, as well as a very effective teaching method. This means that they can be beneficial for both students and researchers. They can aid the actual field work in many ways, whereas they are a very interesting experience for students. They are more effective than common teaching methods and they can aid the understanding of the fundamentals of geology, geomorphology, physical geography, and other geosciences. They are also very



helpful when performing an investigative field work. However, they cannot take the place of actual field trips, as they are not as interactive as the latter, thus rendering them less effective.

**Author contribution**

NE conceived and directed the project. AK, AK, AG, MT, and GS contributed to the design of the virtual fieldtrip. MG and ES contributed to the analysis of the results. NE, AK, AG and MT contributed to the writing of the paper.

**Competing interests**

The authors declare that they have no conflict of interest.

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
