# Peer review of "Interactive virtual fieldtrip as a tool for remote education."

_Geoscience Communication, 2021_

## Referee Comment (RC1)

Review of Paper **Interactive virtual fieldtrip as a tool for remote education.**

Reviewer Denise Balmer

The paper suggests that virtual fieldwork during the Covid-19 pandemic could be used as a tool for examining field study areas and plenty of information is supplied. The paper suggests advantages and disadvantages of virtual against actual fieldwork but does state that virtual fieldwork should not be a substitute for actual fieldwork. However, this paper does not give any specific objectives for the students to study although it suggests under the discussion section that they were looking at geomorphological principles. There is no questionnaire given on which the evaluation discussion is based. It is stated that the students seemed to recommend the idea of Virtual Field Trips although there is no mention of follow up actual field work here.
The idea of virtual fieldwork is useful as a precursor to actual fieldwork, and the paper is interesting from that viewpoint.

I was not able to access the story map at the given point.

I found some difficulty with some of the language use, possibly because of translation issues and have suggested alternatives below. This looks excessive but will add to the ease of reading.
Line 7 and line 30: prosecution to be replaced with execution
Line 31: participants rather than attendants
Line 40: prosecute to be replaced with undertake
Line 45: prosecution replaced with running
Line 63 consequent by subsequent
Line 64 should read alternative TO the field trip
Line 81 should read presented TO students
Line 127: lower or flatter rather than milder
Line 208: should read preventing the moisture penetrating into the room
Line 224: perhaps 'in most coasts' be replaced with 'along most of the shoreline'
Line 229:  should read' artificial beachworks physical properties were LATER studied…
Line 236: should read 'interesting and AIDED the understanding…
Line 273: abilities be replaced with opportunities

An interesting article contrasting Virtual and Actual fieldwork has just appeared in the Teaching Earth Sciences magazine , Vol 46, 1 and 2 for 2021, pp 30 to 33, by Jordan Phethean (j.phethean@derby.ac.uk) which might be of interest.

---

## Author Comment (AC1)

The paper suggests that virtual fieldwork during the Covid-19 pandemic could be used as a tool for examining field study areas and plenty of information is supplied. The paper suggests advantages and disadvantages of virtual against actual fieldwork but does state that virtual fieldwork should not be a substitute for actual fieldwork. However, this paper does not give any specific objectives for the students to study although it suggests under the discussion section that they were looking at geomorphological principles. There is no questionnaire given on which the evaluation discussion is based. It is stated that the students seemed to recommend the idea of Virtual Field Trips although there is no mention of follow up actual field work here.

The goal was two-fold: the virtual fieldtrip was presented to undergraduate students of Geomorphology, and therefore they had the opportunity to "visit" an area that is not within their curricula; so they were able to get acquainted with different geomorphological processes and environments. The virtual fieldtrip was also used in the framework of a series of virtual fieldtrips for Erasmus+ CIVIS. Given that your comment regards to clarify any specific objectives for the students to study, we will try to update this part in our revised manuscript.

Regarding the questionnaire for the evaluation, we can provide the link for the questionnaire used and describe some further information regarding its design and purpose.

The idea of virtual fieldwork is useful as a precursor to actual fieldwork, and the paper is interesting from that viewpoint.

We thank you for the comment and we agree. Virtual fieldtrips can be very useful to prepare students for the actual fieldtrip, so that they may put emphasis on more methodological or mapping issues during the live fieldwork. We will try to highlight this point of view in our revised manuscript.

I was not able to access the story map at the given point.

Not sure why this happened, the link now seems to work fine.

I found some difficulty with some of the language use, possibly because of translation issues and have suggested alternatives below. This looks excessive but will add to the ease of reading.

Thank for the language improvements, we will incorporate them in the revised version of our manuscript.

---

## Author Comment (AC2)

General comments

I do not believe there is not enough original content in this paper for it to be published. The lead author has self-cited throughout and has included much of their own work in Section 3 that is far too detailed and irrelevant for the paper. the The authors have already published results of a similar survey (likely conducted simultaneously as the cohort is the same) in the same Geoscience Communication special issue "Virtual field trips as a tool for indirect geomorphological experience: a case study from the southeastern part of the Gulf of Corinth, Greece". In my opinion, the results presented here should have been included in that publication.

I agree with the points Mohadjer has raised, and, if the authors wish to publish these results, I encourage them to develop this paper considerably in light of the comments below and the suggestions raised by Mohadjer.

We have studied in detail both your comments and the comments posted by Solmaz Mohadjer. We will take into consideration both reviewers suggestions, in order to improve the context and objectives of our manuscript.

Specific comments

24 This is supposition – do you mean field trips can support the classroom-taught principles? I wouldn't say they allow students to understand concepts in any certain terms.

We are not sure we understand the comment. In our point of view, in order for students to better comprehend what they are taught in the class, field trips are a very important part as they can examine sites themselves and see in nature what they sometimes see theoretically through a sketch or a photograph.

40 I'm not convinced this example of the challenges of Greek educators visiting another location is particularly relevant. It's obvious that actual fieldtrips require transportation, time, and resources, it's not necessary to add a weak example.

Thank you for the comment. We will modify this part accordingly.

45 I think you can be satisfied the reader will know what weather conditions are – no need to add examples.

We will remove the examples.

49 Careful with your use of 'paramount significance' here! Is it really? Do you present the evidence?

You are correct. The phrase was somewhat an exaggeration, we will modify this sentence accordingly.

67 I'm confused about your approach and thinking here. Are you testing a hypothesis that virtual field trips are useful supplements but not substitutes to actual field trips? You seem to be suggesting here that you know this already….so why study it?

Well yes, the hypothesis is that virtual field trips are useful supplements but not substitutes and this is why we also included the feedback from our students. We can upgrade this part accordingly.

82 84 of the whole cohort of students completed the questionnaire? Or 84 of the 134 third year students? Created for what purpose? Please add some more information here. What were you asking the students?

The anonymous questionnaire was filled in by 94 students in total. Amongst the aims was for them to evaluate the virtual fieldtrip and how it contributed to their better understanding of geomorphological topics. We will add more details regarding the questionnaire in the methods section and we will also include the link to the questionnaire.

132-133 It is useful to know the reasons why the region was chosen, such as you have done here. Much of the detail in Section 3 is superfluous, and unless it is directly relevant to the aims of the study, I'd advise removing much of it, or explaining why the detail is important to include in this particular paper.

Thank you for this comment. Taking also into consideration a similar comment from Solmaz Mohadjer, we will try to shorten some parts of section 3 and explain better how and why each topic was approached.

Figure 5 Is very poor quality.

We will upgrade this figure, thank you for the comment.

I'm afraid I stopped reviewing at this point as the discussion and conclusions are not original contributions to research (see general comments).

Technical comments

Be consistent with your use of fieldtrip or field trip, fieldwork or field work.

7 I'd advise changing the word prosecution throughout the manuscript. It has two meanings in English.

12 Be consistent throughout with your use of live, real or actual field trip. I would recommend actual.

28 I suggest moving this sentence before the previous and slightly developing the sentence which you provide a reference by Clark. Are you talking here about the power of field trips as shared experiences? If so, please provide a little more information.

31 Advise using students or participants instead of 'attendants'

47 Can be not they are 48 some not most

81 to not at

104 no need to tell us which programme you used to created your map

Thank you for these detailed technical comments, we will make all the suggested changes in our revised manuscript.

---

## Author Comment (AC3)

This is a timely manuscript as the covid pandemic has forced many university programs to provide students access to field studies. I have a couple of questions and comments:

1. What are the study objectives? This was not stated clearly in the text. If the main objective is to compare virtual with real field trips in this specific setting, then the testing of actual field trips to this specific setting is missing. If this is true, then students are asked to compare the virtual trips they have done with a trip that they have not actually done. This needs to be clarified before going into results and discussions.

   Amongst the study objectives was to assess how students understood the topics addressed during the virtual fieldtrip (i.e. coastal evolution, palaeogeographic reconstruction, sea-level changes and geoarchaeology) and how effectively could virtual fieldtrips accomplish that. These students were also asked to complete a small quiz after the end of each topic, which enabled us to get a better idea of how they have better understood some geomorphological processes and landforms. However, we understand the point of view of your comment, and we will try to improve the study objectives in the revised manuscript.

2. I couldn't find the questionnaire used by the authors to evaluate the virtual field trips. Please consider adding it to the main manuscript so that readers can see how the questions were framed. A discussion section on the questionnaire design would be helpful too. For instance, what is the purpose behind each question and how does each question connect to the main objectives of this study. Again, we need to know what those objectives are, so we know what's being evaluated.

   We will add the questionnaire we used as supplementary material.

3. I suggest including the link to the storymap in the manuscript text so that readers can easily access and view it. The hyperlink in line 76 does not work.

   Thank you for the suggestion, we will add it again in the manuscript.

4. I suggest include a workflow and/or a storyboard for the virtual field trips and include these in the method section. This can quickly give readers an overview of the curriculum structure, topics covered, and connections drawn between them. This would be in addition to the snapshots in Fig 1 which at the current resolution are too small to be meaningful.

   Thank you for the suggestion, we will add a workflow in the methods section. We will also improve Figure 1.

5. I think section 3 (study area) needs some context. Since this manuscript is about the virtual field trip, I suggest discussing each subsection (e.g., geodynamics, geology, geomorphology, etc.) in the context of curriculum design and study objectives. For example, when discussing the geomorphology of the study area, it would be useful to know what considerations had to be made to introduce students to geomorphological topics and how does the content covered in this section reflect those considerations. This would make this section much more interesting to readers and more relevant to geoscience communication.

Thank you for the suggestion. We will try to follow your suggestions for section 3 and explain a bit better how each topic was approached.

6. Please include other information about the testing of the curriculum. How long does it take for a student to complete the trip(s)? Did the students complete the trips alone or together with other students? Over how many days the testing was conducted? Did the students have access to their instructor while completing the trips? Did they run into any problems (technical and non-technical)?

Thank you for this comment, we will add such information in the revised manuscript.

7. I agree that virtual and real field trips are two different experiences both for the students and the instructors. However, there are many elements of real trips that can be brought into the virtual world including student-student and teacher-student interactions, data collection and interpretation. Based on my quick review of the content, the virtual field trips used in this study are not designed to be very interactive. Students are often shown images that they can zoom in and out of and sometimes 3D images and videos to check out. The content often reads like an online textbook. One way to make the content more interactive is to ask students to look for specific features in an image or satellite imagery, or show them the features but ask them to identify similar features and/or explain what could these features mean in terms of outcrop history, evolution etc. These are often questions that instructors and students discuss when they are looking at an outcrop in the field, and can be easily brought into virtual trips.

Thank you for this useful comment. In fact, at this point the virtual fieldtrip was not designed to stand alone, but with the presence of the tutor who encourages students to examine landforms and landscapes and comment, discuss on what they mean, how they evolved, etc.